# Evaluation of Small Molecular Polypeptides from the Mantle of *Pinctada Martensii* on Promoting Skin Wound Healing in Mice

**DOI:** 10.3390/molecules24234231

**Published:** 2019-11-21

**Authors:** Faming Yang, Xiaoming Qin, Ting Zhang, Haisheng Lin, Chaohua Zhang

**Affiliations:** 1College of Food Science and Technology, Guangdong Ocean University, Zhanjiang 524088, China; yangfm0123@163.com (F.Y.); ZhTing95@163.com (T.Z.); haishenglin@163.com (H.L.); Zhangch2@139.com (C.Z.); 2National Research and Development Branch Center for Shellfish Processing (Zhanjiang), Zhanjiang 524088, China; 3Guangdong Provincial Key Laboratory of Aquatic Products Processing and Safety, Zhanjiang 524088, China; 4Guangdong Province Engineering Laboratory for Marine Biological Products, Zhanjiang 524088, China; 5Key Laboratory of Advanced Processing of Aquatic Product of Guangdong Higher Education Institution, Zhanjiang 524088, China; 6South China Sea Bio-Resource Exploitation and Utilization Collaborative Innovation Center, Zhanjiang 524088, China

**Keywords:** *Pinctada martensii*, mantle, small molecule peptide, skin, wound healing

## Abstract

Skin wound healing, especially chronic wound healing, is a common challenging clinical problem. It is urgent to broaden the sources of bioactive substances that can safely and efficiently promote skin wound healing. This study aimed to observe the effects of small molecular peptides (SMPs) of the mantle of *Pinctada martensii* on wound healing. After physicochemical analysis of amino acids and mass spectrometry of SMPs, the effect of SMPs on promoting healing was studied through a whole cortex wound model on the back of mice for 18 consecutive days. The results showed that SMPs consisted of polypeptides with a molecular weight of 302.17–2936.43 Da. The content of polypeptides containing 2–15 amino acids accounted for 73.87%, and the hydrophobic amino acids accounted for 56.51%. Results of in vitro experimentation showed that SMPs possess a procoagulant effect, but no antibacterial activity. Results of in vivo experiments indicated that SMPs inhibit inflammatory response by secretion of anti-inflammatory factor IL-10 during the inflammatory phase; during the proliferative phase, SMPs promote the proliferation of fibroblasts and keratinocytes. The secretion of transforming growth factor-β1 and cyclin D1 accelerates the epithelialization and contraction of wounds. In the proliferative phase, SMPs effectively promote collagen deposition and partially inhibit superficial scar hyperplasia. These results show that SMPs promotes dermal wound healing in mice and have a tremendous potential for development and utilization in skin wound healing.

## 1. Introduction

Skin covers the whole body and protects various tissues and organs from physical, mechanical, chemical, and pathogenic microorganisms [1,2]. Skin wound healing is a dynamic process that coordinates a series of complex processes involved in tissue repair, including hemostasis, inflammation, cell proliferation, and remodeling. The recovery of the vascular system, reduced inflammation, promotion of fibroblast differentiation, and connective tissue formation are of great significance in wound healing [3,4]. Skin damage, especially chronic wounds, burns, and skin wound infections require painstaking long-term treatment, which imposes a significant economic burden on the health care system worldwide. Additionally, the aging population, along with the rise in diabetes and obesity, continues to increase the prevalence of chronic wounds [5]. Although the combination of natural products, such as the Chinese herbal extract ointment, provides a new way to treat wounds, traditional products used for wound treatment such as bandages and gauze cannot be completely replaced [6]. Moreover, natural scar-free formulations for skin trauma and clinical nutrition for trauma patients are scarce. Thus, these limitations of the existing skin injury treatments prompt further exploration of a harmless, feasible, and cost-effective formulation for skin injury repair. It is imperative to identify biologically active substances that can be used for wound healing. In-depth research on the functions of marine-derived bioactive peptides has revealed that some bioactive peptides have a high potential for application in skin tissue wound healing, such as marine collagen peptides and fish skin collagen peptides [7,8]. However, there are no reports on the application of bioactive peptide substances derived from shellfish in wound healing.

*Pinctada martensii* was first used for industrialized pearl production in Japan in the late 1890s [9]. It is famous for its ability to produce high-quality pearls and accounts for more than 90% of seawater pearls. It is the main shellfish that is used for cultivating pearls in coastal areas of southern China [10,11]. Cultured pearl oyster is mainly used for collecting pearls. The meat of pearl oyster as a by-product is mainly used as fresh food or simply processed into animal feed. It gives low output and is an environmental pollutant. Therefore, high-value utilization is an important means to improve the economic value of the *Pinctada martensii* industry. Studies have shown that the main nutrient component of the *Pinctada martensii* mantle is protein [11], as the tissue encapsulating the nucleus of pearl for artificial “nuclear insertion surgery” has strong cellular proliferation and tissue repair ability. Thus, it plays an important role in the formation of pearls [12]. Although the angiotensin-I-converting enzyme (ACE) inhibitory peptides [11] can be obtained by enzymatic hydrolysis of *Pinctada martensii* meat. However, there are no reports on wound healing activity of *Pinctada martensii*.

This study describes the preparation of small molecular polypeptides (SMPs) from the mantle of *Pinctada martensii,* and evaluates the effect of daubing on skin wound healing in male Kunming mice, which provides a theoretical basis for the development and clinical application of related products.

## 2. Results

### 2.1. Mass Spectrometry Analysis of Molecular Weight Distribution of SMPs

Figure 1 shows the total ion mass spectrum of SMPs, as determined by primary mass spectrometry, along with the overall molecular weight distribution. Next, the uniprot-Pinctada fucata database was searched using Maxquant (1.6.2.10), which identified a total of 1328 proteins in the molecular weight range of 302.17–2936.43 Da.

### 2.2. Amino Acid Composition of SMPs

The amino acid composition and content (g/100 g) of SMPs are shown in Table 1. Among the 17 amino acids found in SMPs, glutamic acid (5.66 g/100 g), alanine (2.71 g/100 g), glycine (2.015 g/100 g), leucine (2.07 g/100 g), and phenylalanine (1.70 g/100 g). Essential amino acids (excluding tryptophan) accounted for 33.77% of the total amino acids. Among them, essential amino acids such as glutamic acid and proline (except tryptophan), and branched-chain amino acids, such as leucine and isoleucine, accounted for 21.83% and 18.89% of the total amino acids, respectively. In this study, hydrophobic amino acids accounted for 56.51% of the total amino acids, and hydrophilic amino acids accounted for 31.3% of the total amino acids.

### 2.3. Characteristic Peptide Sequence Analysis of SMPs

Peptide fingerprinting of 21 characteristic peptides in SMPs was analyzed using a liquid chromatography-mass spectrometry (LC-MS/MS). The molecular weight of SMPs were in the range 1512.83 Da to 2241.05 Da (amino acid residue 10-21) (Table 2). Two peptide fragments of Gln-Leu and Asp-Leu recurred in the 21 characteristic peptide sequences of SMPs, and Leu recurred at the beginning of the characteristic peptide, while Asn and Arg reappeared at the tail of the characteristic peptide. Additionally, Val, Leu, and Ile were found in the middle of the SMPs characteristic peptides.

### 2.4. Wound Healing In Vitro

#### 2.4.1. Effect of SMPs on Coagulation In Vitro

The effect of SMPs on plasma recalcification coagulation time is shown in Table 3. Compared with the negative control group, the positive control group and the SMPs group significantly shortened the plasma recalcification time (*p* < 0.05), and the shorter the time of plasma recalcification, the better the procoagulant effect. The results indicated that SMPs exerted a certain amount of procoagulant effect, which was evaluated in subsequent experiments.

#### 2.4.2. Antibacterial Activity of SMPs

The antibacterial effect of SMPs against 13 types of common bacteria found in the living environment of pearl oyster and wound infection: *Staphyloccocus aureus*, *Listeria monocytogenes, Streptococcus iniae, Lactococcusgarviea, Streptococcus agalactiae, Bacillus, Escherichia coli, Pseudomonas aeruginosa, Vibrio alginolyticus, Vibrio Parahemolyticus, Shewanella alga, Vibrio harveyi, Aeromonas hydrophila.* They were tested in vitro by the micro-radial diffusion method and half dilution method. The results showed that SMPs possessed no antibacterial activity.

### 2.5. Wound Healing In Vivo

#### 2.5.1. Effects of SMPs on Wound Healing

Figure 2 presents the effect of SMPs in a wound healing model in mice. Generally, the percentage of wound closure in each group increased with time. Wound healing images revealed that the healing of the surface and skin endothelium were accelerated after treatment with SMPs (Figure 2A). Overall, the healing effect of the SMPs group was more significant than the control group, and no hypertrophic scars and keloids were formed. As shown in Figure 2B, there was no significant difference in the healing rate between the groups (*p* > 0.05), but the healing rate of the SMPs group was higher than that of the control group six to eight days after model establishment. However, after 10–14 days, the percentage of wound closure of the SMPs group was significantly higher than that of the control group (*p* < 0.01). There was no significant difference in healing rate between the groups on days 16–18 (*p* > 0.05).

As shown in Table 4, the eschar fading time of the wounds in the SMPs group was shorter than that of the control group (*p* > 0.05). Although there was no significant difference in epithelialization time between the groups, the epithelialization time was the shortest in the SMPs group.

#### 2.5.2. Effects of SMPs on Cytokine

As depicted in Figure 3, from day 3 to day 5, the IL-6 content of the control group showed no significant change, however, it showed a significant increase in the SMPs group. Notably, on day 3, the IL-6 content of the SMPs group was significantly lower than that of the control group (*p* < 0.01). Meanwhile, on day 5, the IL-10 content in the negative control group and the SMPs group increased significantly compared with day 3. Moreover, on day 3, the SMPs group significantly promoted IL-10 secretion compared with the control group (*p* < 0.05).

#### 2.5.3. Effects of SMPs on Growth Factors

As shown in Table 5, compared with the negative control group, there was a significant increase in the levels of TGF-β1 in the skin of the mice that were administrated SMPs (*p* < 0.05). Although, compared with the negative control group, SMPs could promote an increase in CCND1 content, there was no significant difference in the levels of CCND1 in the SMPs group as compared with both the controls (*p* > 0.05).

#### 2.5.4. Histological Evaluation

The effect of SMPs on skin wound healing in mice was studied microscopically using H&E staining in Figure 4. On day 3 of the experiment, in the negative control group, the positive control group and the SMPs group showed a weaker inflammatory response. On day 7 of the experiment, there were still few inflammatory cells in the negative control group. The epidermal layer of the wound was incompletely healed and the collagen fibers in the dermis were sparse and few. However, in the positive control group and SMPs group, no inflammatory reaction was observed, and the wound formed a coherent epidermis, and more granulation tissue formed in the dermis. On day 18 of the experiment, compared with the negative control group, the repair of epidermis and dermis in the positive control group and SMPs group resembled normal skin.

#### 2.5.5. SMPs Promote the Expression of EGF, FGF, and CD31

As shown in Figure 5, SMPs promoted the expression of CD31, FGF, and EGF compared with the control group. On day 7, SMPs showed significant differences in promoting CD31 and FGF expression (*p* < 0.05). On day 18, the experimental groups showed no significant difference in the expression of CD31 and EGF, whereas, the SMPs group showed significant upregulation of FGF (*p* < 0.05).

#### 2.5.6. Effects of SMPs on Collagen Bundles and Scar Residual Rate

The SMPs group showed more organized collagen bundles and collagen deposition than the control group with Masson stained sections. The SMPs group shows an improved epithelial formation phase in Figure 6A. These results suggested that SMPs could significantly promote the secretion and deposition of collagen and achieve better skin wound healing in mice. The results of scar reduction showed that SMPs could effectively inhibit superficial scar reduction (*p* < 0.05), as shown in Figure 6B.

## 3. Discussion

Wound healing is a dynamic coordinated tissue repair process that consists of three/four overlapping stages: Hemorrhage/inflammation, cell proliferation, and tissue remodeling [13]. This study carried out in vitro/in vivo experiments to explore the effect of topical SMPs on skin wound healing in mice. This study was performed to determine the pathological effects and biochemical indicators associated with the invasive tissue in experimental animals at various stages, and to explore the possible mechanisms of wound healing. It provided a theoretical as well as a research basis for future in-depth exploration of mechanism.

Nutrition is an important part of wound healing, and nutritional deficiencies will delay wound healing in patients [14]. Nutritional supplements are generally absorbed by the wound on oral administration [15]. Therefore, the method of applying SMPs may limit the absorption of macromolecular nutrients, such as proteins, but smear administration would hydrate the stratum corneum, increase permeability, and enhance drug absorption [16]. At the same time, small peptides, trace elements, and other bioactive substances directly act on the skin wounds, thus shortening the reaction time and accelerating the healing process through smearing [16]. In this study, hydrophobic amino acids accounted for 56.51% of the total amino acids, and hydrophilic amino acids accounted for 31.3% of the total amino acids. Another study found that hydrophobic amino acids were closely related to immunological activity [17]. At the same time, another study found that the mantle of *Pinctada martensii* contains a relatively rich total carotenoid content, which is also related to immunity [18]. This provides a nutritional and theoretical basis for SMPs to improve immunity during the inflammatory phase of skin wound healing and to produce an inflammatory response. At the same time, during wound healing, proteins are being continuously synthesized, thus close attention towards nitrogen balance is required. SMPs, which are rich in glutamic acid, glycine (in pearl oyster mantle edge tissue, the most highly expressed genes had glycine-rich protein [18]), cysteine, and phenylalanine can achieve this nitrogen balance that is beneficial to angiogenesis, fibroblast proliferation, collagen synthesis, and wound remodeling [19]. Normal health and physiological functions require at least 20 vitamins and at least 16 minerals and trace elements [14]. In nutrition, it provides a theoretical basis for the wound healing potential of SMPs.

SMPs possess biological activity such as good membrane-permeability and cellular activation, [20]. The peptide sequences including Gln-Leu and Asp-Leu are repeatedly present in the SMPs feature peptides (Table 2). Studies have shown that Gln-Leu and Asp-Leu play an important role in the removal of free radicals in peptides [21,22,23]. The first amino acid of the characteristic peptide is generally Leu. Arg often appears at the end of the characteristic peptide, and Asp often appears in the middle of the feature peptide. Another study found that arginine, a non-essential amino acid, is a necessary matrix component for certain conditions in highly stressed adults. Therefore, Glu-Leu and Ala-Arg play an important role in bioactive peptides that promote wound healing [24]. Other studies have also shown that arginine supplementation enhances collagen deposition [25,26]. Additionally, SMPs with small molecular weights (<3000 Da) can be quickly absorbed and provide a nutritional matrix for wound healing.

Hemostasis is a critical step in emergency medical care [27]. After skin injury, platelets are stimulated to gather with fibrin-forming clots to quickly and effectively stop bleeding, and is extremely important for surgical and trauma emergencies, especially in traumatic battlefields [28]. Fast blood coagulation during the early stages of the coagulation cascade reflects fast and efficient hemostasis [29]. The time of plasma calcium recovery refers to the time required for the formation of clots after the whole blood is decalcified or the plasma is added with calcium. It has the same significance as the time of whole blood coagulation, and its sensitivity is higher than that of whole blood coagulation. In vitro procoagulant studies have shown that SMPs can accelerate blood clotting by shortening the time of plasma complex calcium to achieve rapid hemostasis, which also enriches the source of hemostatic agents. However, the further mechanism of the procoagulant effect of SMPs is not clear, and further research is needed. Additionally, antibacterial agents have a great influence on the speed and quality of skin wound healing. If the healing rate is too slow, it will easily cause infection and can be life-threatening. This study showed that SMPs do not possess antibacterial activity in vitro. Thus, SMPs can be mixed with antibacterial bioactive substances to further enhance its healing function, for the development of relevant biomedicine.

Inflammation results from complex interactions between soluble cytokines such as IL-6 and IL-10 and cells, and after injury, the tissue-resident macrophages immediately trigger a local inflammatory response [30]. When inflammation subsides, neutrophils, and associated molecular factors are reduced, leading to changes in the microenvironment of the tissues [31]. Thus, the inflammatory macrophage phenotype was transformed into an anti-inflammatory cell type that directly promotes fibrosis during granulation tissue formation [32]. This decrease in inflammation is essential to prevent the development of the chronic healing process, which eventually leads to the formation of fatty scars [13]. Macrophages have extensive surface markers, which are divided into two subsets: M1 and M2 macrophages. IL-6 secreted by M1 macrophages is a pro-inflammatory cytokine, and IL-10 secreted by M2 macrophages is an anti-inflammatory cytokine [33]. On day 3 of the experiment, SMPs promoted the secretion of IL-10, and inhibited the secretion of IL-6. On day 5, there was no significant difference in the levels of IL-6 in each group, and IL-10 secretion in the positive control group was significantly lower than other groups. Thus, the anti-inflammatory cytokine IL-10, which was significantly upregulated throughout the study period, indicated that the SMPs group healed better [34]. The results of the H&E staining show that the SMPs group did not have significant inflammatory cell invasion, consistent with the results of SMPs with enhanced immunity. These results suggested that SMPs promote the proliferation of M2 macrophages and inhibit the inflammatory response.

Keratinocyte and fibroblasts play a vital role in wound healing [35]. Moreover, the wound healing process requires the structural and functional reconstruction and contraction of differentiated muscle fibroblasts in granulation [36]. Day 7 of H&E staining, and day 7 and day 18 of immunohistochemical microscopy showed that SMPs could effectively promote the proliferation of keratinocytes and fibroblasts during healing, thereby accelerating epithelialization and production of skin appendage, thus achieving high-quality healing. Growth factors play an important role in the complex process of wound healing and tissue regeneration by forming the ideal environment conducive to the skin wound healing process. Inactive TGF-β1 is secreted by platelets, macrophages, fibroblasts, and other cells, which gets activated in an acidic environment in the body at fractures and healing wounds, thus promoting the growth of fibroblasts and extracellular matrix (ECM) components such as collagen [37]. The main function of CCND1 is to promote cellular proliferation [38]. On day 7, ELISA resultes showed that the SMPs significantly enhanced the secretion of TGF-β1 and effected the secretion of CCND1. These results suggested that the SMPs may accelerate trauma repair by accelerating the expression of TGF-β1 and CCND1. Meanwhile, re-epithelialization results in the recovery of the vascular network [39]. CD31 stained endothelial cells, macrophages, platelets reflect a larger vascular orientation, and EGF plays a crucial role in platelet survival [40]. The results of immunohistochemistry showed that SMPs could significantly promote the expression of CD31 and EGF, thus promoting angiogenesis and wound healing.

The final stage of skin wound healing, the tissue remodeling phase, results in the formation of scars. The scar formation process involves two main phenomena: Re-epithelialization and maturation of granulation tissue [41]. The tissue remodeling phase can last for a long time. At this stage, the matrix cells coordinate the renewal of extracellular molecules in the wound bed, resulting in increased tissue strength, and collagen deposition, which plays an important role in wound remodeling [42]. According to the day 18 results of H&E staining, the SMPs group significantly accelerated skin contraction, skin organ maturation, and wound healing. Masson’s trichrome staining is commonly used to identify collagen in wounds [43]. Masson’s trichrome stain dyes collagen blue. Thus, a deep blue color in the wound indicates more collagen deposition [44]. This study showed an eddy blue color after treatment with SMPs, compared to the control group. Thus, it showed that SMPs treatment enhances collagen deposition and re-epithelialization, further accelerating wound healing, and inhibiting scar formation. This was verified based on the results of scar healing rate. Moreover, TGF-β1 also promotes ECM deposition along with activating and inducing fibroblasts to secrete type I collagen [45]. IL-10 may reduce scar formation during remodeling by reducing the recruitment of inflammatory cells [46]. Therefore, the residual scar formation along with the formation of collagen bundles in the wound area may be attributed to the overexpression of IL-10 (based on images). Based on the above results, SMPs treatment can promote collagen deposition and re-epithelialization, thus accelerating wound healing. In this study, due to the short test period and the type of collagen in the skin on the 18th day, there is no more clear and in-depth explanation of the mechanism of SMPs on the formation of collagen in the wound and the inhibition of scar formation, which needs to be optimized in the subsequent experiments and further mechanism research.

## 4. Materials and Methods

### 4.1. Materials

*Pinctada martensii* meat was collected from the Liusha farm in Leizhou City, Zhanjiang China in May 2019. Yunnan Baiyao (purchased in Zhanjiang China) was purchased from Yunnan Baiyao Group. SPF male mouse with bodyweight 20 ± 2 g was purchased from Pengyue Experimental Animal Breeding Co., Ltd. (Jinan, China) and production license No. SCXK (Lu) 20190003. The Laboratory license number SYXK (Guangdong) 2019-0204, Animal Laboratory, Guangdong Ocean University; Neutral Protease (3 × 10^5^ Ug^−1^) was purchased from Pangbo Biological Engineering Co., Ltd. (Nanning, China), IL-6 (Interleukin-6), IL-10 (Interleukin-10), TGF-β1 (Transforming Growth factor β1), Cyclin D1 were purchased from Nanjing Jiancheng Bioengineering Institute. The strain required for the antibacterial test was provided by the Guangdong Provincial Key Laboratory of Aquatic Products Processing and Safety. Other required reagents were either analytically pure or chromatographically pure.

### 4.2. Preparation of SMPs

The mantle was washed, drained, and water was added to the ratio of 1:3 (*m*/*v*, mantle: water), followed by the addition of neutral protease at 1000 U/g (raw material). The hydrolysis reaction was performed at 53 °C for 5 h. Subsequently, the enzyme was inactivated at 100 °C for 10 min followed by centrifugation at 8000 rpm at 4 °C to obtain the supernatant [8]. The macromolecules and particulate impurities were removed through a 200 μm ceramic membrane microfiltration device, and the SMPs were obtained through the <3 kDa ultrafiltration system. The samples were freeze-dried to prepare a lyophilized powder for future experimentation.

### 4.3. Determination of Molecular Weight Distribution

SMPs were subjected to reductive alkylation treatment: 100 μL/tube of SMPs were mixed with 10 mmol/L dithiothreitol in a water bath at 56 °C for 1 h, 55 mmol/L iodoacetamide was added at a concentration of 100 μL/tube and the reaction was carried out in the dark at room temperature for 1 h. Next, the mixture was desalted using a self-priming desalting column, and the solvent was evaporated in a vacuum centrifuge at 45 °C. The LTQ VELOS ESI cation calibration solution was used for molecular weight distribution analysis by liquid chromatography-mass spectrometry (LC-MS/MS). The formulation was: Caffeine (2 μg/mL), MRFA (1 μg/mL), Ultramark 1621 (0.001%), and n-butylamine (0.000 5%) in acetonitrile (50%), aqueous methanol (25%) and acetic acid (1%).

### 4.4. Amino Acid Composition of SMPs

The SMPs were hydrolyzed using 6 mol/L HCl [8]. The solution was analyzed with an amino acid analyzer (L-8900, Hitachi, Tokyo, Japan).

### 4.5. Major Peptide Sequence Analysis of SMPs

SMPs were identified using Thermo Q Exactive™ Hybrid Quadrupole-Orbitrap™ Mass Spectrometer with electrospray ionization (ESI) interface (Thermo Fisher Scientif, Waltham, MA, USA). The secondary mass spectrometry was based on the results of SMPs primary mass spectrometer total ion map. The SMPs analysis was performed on mass spectrometer set from 350 to 1800 *m*/*z*. The mass spectrometer was operated under the following conditions: Resolution: 75,000, AGCtarget: 1e5, MaximumIT: 60 ms, TopN: 20, NCE/steppedNCE: 27.

### 4.6. In Vitro Antibacterial Test

The antibacterial activity was verified by a micro-radial diffusion method and half dilution method [47].

### 4.7. In Vitro Plasma Recalcification Time Test

Fresh rabbit blood with sodium citrate as anticoagulant was centrifuged at 1000 rpm for 10 min, and plasma was separated for further use. Plasma (0.1 mL) was added to each tube, followed by the addition of 0.1 mL of SMPs at a concentration of 1 mg/mL, 2% citric acid solution as a negative control, and Yunnan Baiyao (1 mg/mL) as a positive control, and incubated in a 37 °C water bath for 1 min. Next, 0.1 mL of 0.025 mol/L calcium chloride solution was added and placed in a 37 °C water bath. At the same time, we started the stopwatch and slowly tilted the tube every 15 s until the white granular fibrin appeared. Next, we recorded the time taken for recalcification. The experiment was repeated five times for each sample and the average was taken [48].

### 4.8. Animal Grouping and Establishment of Trauma Model

The mice were randomly divided into three groups, 19 mice per group, including negative control group, positive control group (The Chinese herbal medicine that was used as the positive control is called Yunnan Baiyao, which is made of precious medicinal materials Puhuang, Baiji, etc. It relieves blood stasis, promotes blood circulation, relieves pain, detoxification, and swelling; dose: 2–3 mg/day, administration method: Topical use), and SMPs group (made into a paste, administration method: Topical use, dose: 3–5 mg/day). After the mice were anesthetized with pentobarbital sodium (50 mg/kg), their back was shaved and disinfected, and the whole layer of skin with a diameter of 0.8 cm was cut off creating a full cortical wound animal model. The mice were housed in cages and administered daily. The wounds were observed and photographed every two days, the wound size was measured, and the results were recorded. Each set of experiments was performed in triplicates. The study was approved by the Guangdong Ocean University (Zhanjiang, China) Experimental Animal Care Ethics Committee (Approval No.: 20190001, Approval Date: 17 June, 2019). All experiments were in accordance with the ARRIVE guidelines and were conducted in accordance with the National Institutes of Health guidelines for the care and use of laboratory animals (NIH Publication 8023, revised 1978).

### 4.9. Percentage of Wound Closure and Scar Residual Rate

The wound diameter (cm) was measured once every two days with a vernier caliper to calculate the percentage of wound closure:
Percentage of wound closure = (D_0_−D_n_)/D_0 _× 100(1)
Scar residual rate = (D_0_−D_n_)/D_0 _× 100%(2)
where D_0_ and Dn are the diameters of the initial and unhealed wound-shaped or the diameters of the initial wound shape and the shape of the scar after healing, respectively, in cm.

### 4.10. Histological Assessment

After the mice were sacrificed, the tissues were taken for histological analysis. The samples were fixed in 4% paraformaldehyde at 4 °C for 24 h, followed by dehydration by immersing in a 70–100% ethanol solution. According to the staining procedure, the skin tissues of mice were stained with H&E and Masson’s staining, respectively, and histopathological analysis was performed using microscope (Olympus IX51, Tokyo, Japanese).

### 4.11. Immunohistochemistry

We used immunohistochemical techniques to evaluate the expression of EGF (epidermal growth factor), FGF (fibroblast growth factor), and CD31 (platelet endothelial cell adhesion molecule-1, PECAM-1). After the tissue sections were dewaxed and hydrated and the hydrothermal antigen was repaired, the enzyme was inactivated using a 3% H_2_O_2_ -methanol solution, and the antigen was recovered with a citrate buffer (pH 6.0). Subsequently, 50–100 µL of ready-to-use goat serum was added dropwise, followed by incubation at room temperature for 20 min. Next, 50–100 µL of primary antibody (diluted 1:200) was added dropwise, and the mixture was incubated at 37 °C for 2 h in a wet box. A universal IgG antibody-Fab fragment-HRP multimer (50 µL) was added dropwise, and the mixture was incubated at 37 °C for 30 min at room temperature and washed three times with PBS. After color development, the slides were evaluated under an optical microscope, and each photo was analyzed using Image-pro plus 6.0 (Media Cybernetics, Inc., Rockville, MD, USA) software to obtain positive cumulative light density.

### 4.12. Elisa Measures Inflammatory Factors and Growth Factors

After centrifugation at 10,000 rpm for 15 min at 4 °C, the supernatant of the homogenized tissue was collected as a protein extract. Quantitative analysis of IL-6, IL-10, TGF-β1, and CCND1 in 10% (*v*/*v*) back skin homogenate supernatant was performed using an ELISA kit (Nanjing Jiancheng, Nanjing, China) [49].

### 4.13. Statistical Analysis

The experimental data were expressed as mean ± standard deviation (mean ± S.D.). One-way analysis of variance was performed using SPSS20 software (IBM, Armonk, NY, USA). Multiple comparisons between groups were performed by the LSD method, p-values of less than 0.05 were considered to be statistically significant. The experimental images were processed using Image J (National Institutes of Health, Bethesda, Maryland, USA).

## 5. Conclusions

In this study, SMPs extracted from the mantle of *Pinctada martensii* consisted of polypeptides with molecular weights less than 3 kDa and had characteristic peptides sequences. In vitro and in vivo tests have shown that SMPs possesses certain procoagulant activity, has the ability to enhance the percentage of wound closure, shorten epithelialization time, inhibit inflammatory response, promote secretion of fibroblasts and keratinocytes, accelerate collagen cross-linking and deposition. Thus, promoting wound healing. The expression of key cytokines and growth factors confirmed the role of SMPs in promoting skin wound healing. Future studies will explore the specific molecular mechanisms involved in wound healing.

## Figures and Tables

**Figure 1 molecules-24-04231-f001:**
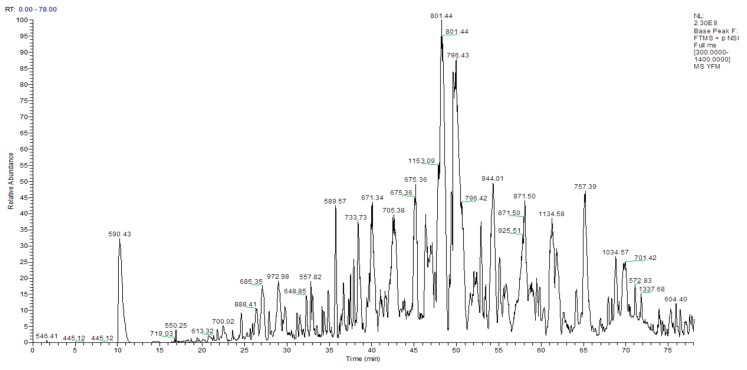
Total ion chromatogram of small molecular peptides (SMPs).

**Figure 2 molecules-24-04231-f002:**
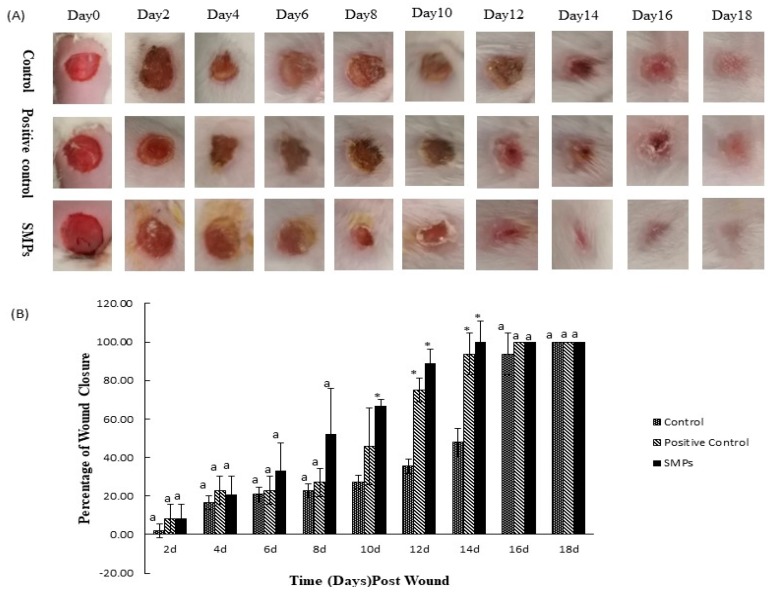
Effect of SMPs on wound healing. (**A**) Representative photos of each group of mice. (**B**) percentage of wound closure (calculated once every two days). Different superscript letters on the same day indicated significant differences between the groups (*p* < 0.05) and insignificant different (*p* > 0.05), respectively, * indicates a very significant difference (*p* < 0.01).

**Figure 3 molecules-24-04231-f003:**
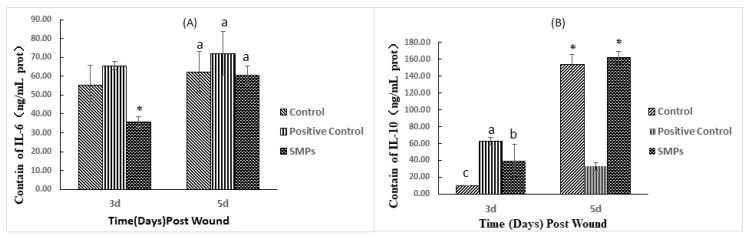
Effect of SMPs on inflammatory factors in wound skin tissue. (**A**) The content of IL-6 in the experimental group mice at three and five days after modeling. (**B**) IL-10 content in mice of the experimental group at three and five days after modeling. Different and same letters within the same row indicate significant (*p < 0.05*) and insignificant different (*p* > 0.05), respectively, * indicates a very significant difference (*p* < 0.01).

**Figure 4 molecules-24-04231-f004:**
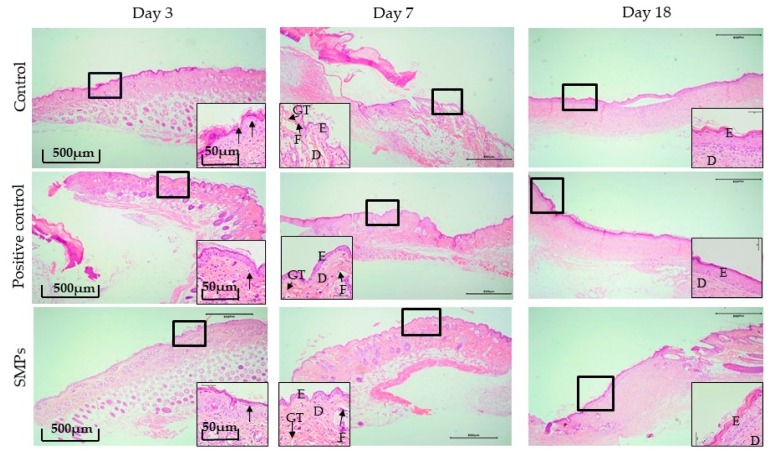
H&E stain histological analysis (4X). Note: The black bold frame indicates the wound site, and the picture at the bottom corner of the picture shows a magnified image of the wound (40X). On the third day, black arrows indicate inflammatory cell infiltration. The letters **D**, **E**, **F**, and **GT** represent the dermis layer, the epidermal layer, fibroblasts, and granulation tissue, respectively.

**Figure 5 molecules-24-04231-f005:**
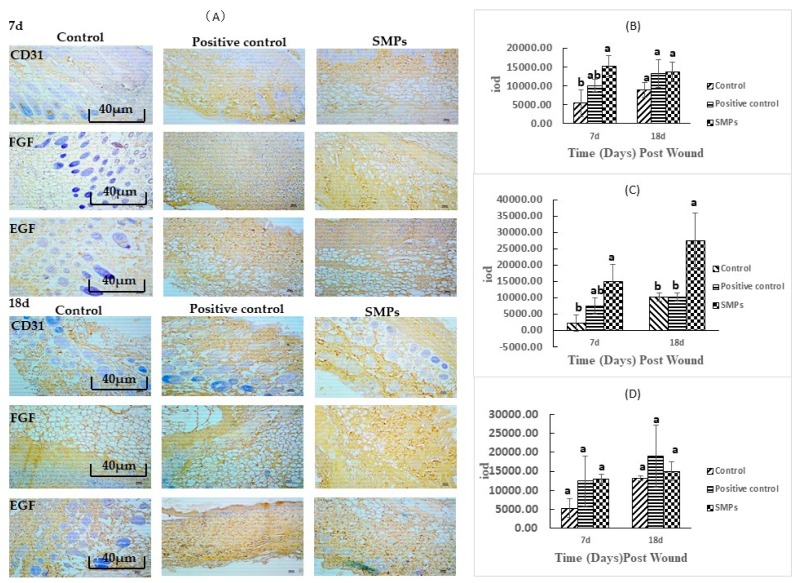
Immunohistochemical analysis chart. (**A**) Representative images of CD31, FGF and EGF immunostaining on days 7 and 18 (scale bar: 40 μm). (**B**) Expression of CD31 at 7d and 18d after trauma. (**C**) Expression of FGF at 7d and 18d after trauma. (**D**) Expression of EGF at 7d and 18d after trauma. Different and same letters within the same row indicate significant (*p* < 0.05) and insignificant different (*p* > 0.05), respectively.

**Figure 6 molecules-24-04231-f006:**
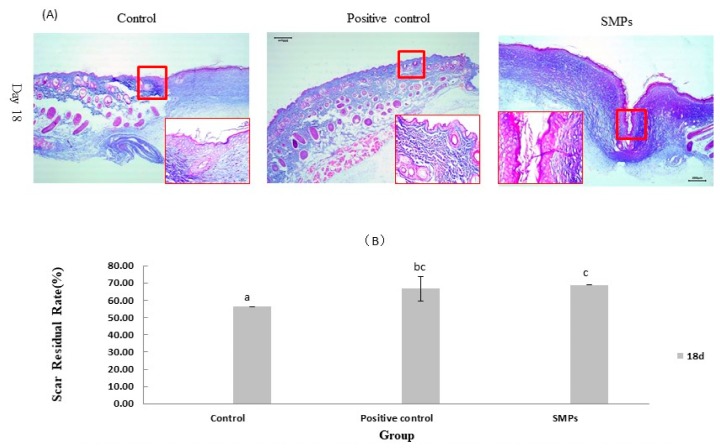
Effects of SMPs on collagen bundles and epithelialization. (**A**) Representative images of Masson stained wound sections (10× and 40×). Sliced blue indicates collagen, red indicates cytoplasm and muscle fibers. (**B**) Scar residual rate. Different and same letters within the same row indicate significant (*p* < 0.05) and insignificant different (*p* > 0.05), respectively.

**Table 1 molecules-24-04231-t001:** Composition and contents of amino acids of SMPs.

Amino Acid	Contains (g/100 g)	Amino Acid	Contains (g/100 g)
Aspartic acid	1.94	Methionine *	0.74
Threonine *	0.90	Isoleucine *	1.36
Serine	0.62	Leucine *	2.07
Glutamic acid	5.66	Tyrosine	0.34
Proline	1.35	Phenylalanine *	1.28
Glycine	2.15	Lysine	1.94
Alanine	2.71	Histidine	0.32
Cysteine	1.17	Arginine *	1.17
Valine *	1.76	Total	27.48

Note: * essential amino acid. Because of the acid hydrolysis and decomposition, the method cannot be determined tryptophan.

**Table 2 molecules-24-04231-t002:** Main peptide sequences analysis from of SMPs.

Sequence	Peptide Sequence of SMPs	Molecular Mass (Da)	Score
1	Ser-Leu-Met-Asp-Thr-Asp-Val-Ser-Thr-Lys-Val-Val-Arg-Asn	1890.97	293.31
2	Leu-Ala-Lys-Ala-Lys-Ser-Gln-Leu-Gln-Leu-Gln-Leu-Asp-Asp-Leu-Lys-Arg-Asn	2081.20	245.4
3	Leu-Lys-Ala-Gln-Val-Asp-Asp-Leu-Thr-Arg-Gln-Leu-Asn-Asp-Ala-Asn-Asn	1926.98	238.43
4	Ala-Gln-Val-Asp-Asp-Leu-Thr-Arg-Gln-Leu-Asn-Asp-Ala-Asn-Asn-Ala-Lys	1884.93	228.38
5	Val-Leu-Asp-Cys-His-Thr-Ala-His-Ile-Ala-Cys-Lys	1423.67	226.09
6	Val-Met-Val-Gly-Met-Gly-Gln-Lys-Asp-Ser-Tyr-Val-Gly-Asp-Glu-Ala-Gln-Ser-Lys-Arg-Gly	2241.05	222.45
7	Leu-Phe-Glu-Asp-Thr-Asn-Leu-Cys-Ala-Ile-His-Ala-Lys-Arg	1686.85	220.94
8	Leu-Ile-Glu-Glu-Ala-Glu-His-Arg-Ala-Asp-Asn-Ala-Glu-Lys-Asn	1737.83	220.37
9	Leu-Ala-Lys-Ala-Lys-Ser-Gln-Leu-Gln-Leu-Gln-Leu-Asp-Asp-Leu-Lys-Arg	1967.15	216.39
10	Ile-Ala-Ala-Met-Gln-Ser-Asp-Leu-Asp-Asp-Ala-Leu-Asn-Ala-Gln-Arg	1730.83	216.03
11	Leu-Thr-Asp-Asp-Gln-Val-Asp-Glu-Ile-Ile-Arg-Asn-Thr-Asp-Leu-Gln-Glu-Asp	2130.99	214.48
12	Ser-His-Ser-Gly-Phe-Pho-Phe-Pho-Pro-Cys	1131.48	213.54
13	Ala-His-Ala-Glu-Ser-Lys-Val-Asp-Ala-Leu-Glu-Gly-Ser-Leu-Ala-Arg	1652.85	211.38
14	Leu-Thr-Gln-Glu-Asn-Phe-Asp-Leu-Gln-His-Gln-Val-Gln-Glu	1727.81	207.34
15	Met-Gln-Ser-Asp-Leu-Asp-Asp-Ala-Leu-Asn-Ala-Gln-Arg	1475.67	207.34
16	Ser-Leu-Met-Asp-Leu-Asp-Thr-Asp-Val-Ser-Thr-Lys-Val-Val-Lys	1677.86	206.11
17	Leu-Asp-Thr-Asp-Val-Ser-Thr-Lys-Val-Val-Arg-Val-Asn	1444.79	203.02
18	Ser-Gln-Leu-Gln-Leu-Gln-Leu-Asp-Asp-Leu-Lys-Arg-Asn	1569.85	201.75
19	Leu-Lys-Ala-Gln-Val-Asp-Asp-Leu-Thr-Arg-Gln-Leu-Asn-Asp-Ala-Asn-Asn-Ala-Lys-Ala	2197.14	199.58
20	Val-Arg-Ile-Gln-Glu-Leu-Glu-Asp-Leu-Leu-Glu-Gln-Gln-Arg	1767.95	197.38
21	Leu-Lys-Ala-Gln-Val-Asp-Asp-Leu-Thr-Arg-Gln-Leu-Asn	1512.83	197.31

Note: Scores obtained by scoring a known protein database to measure the similarity between theoretical mass spectra and experimental mass spectra.

**Table 3 molecules-24-04231-t003:** Determination of plasma recalcification coagulation time.

Group	Clotting Time(s)
2% citric acid	162.80 ± 3.96 ^a^
Yunnan Baiyao	81.20 ± 1.10 ^c^
SMPs	119.80 ± 0.84 ^b^

Note: Different and same letters within the same row indicate significant (*p* < 0.05) and insignificant different (*p* > 0.05), respectively.

**Table 4 molecules-24-04231-t004:** Effect of SMPs on wound loss and epithelialization (time mean ± S.D., Day).

Group	Dislocation Time (Day)	Epithelialization Time (Day)
Control group	12.00 ± 1.73	16.67 ± 1.15 ^a^
Positive control group	12.00 ± 0.00	15.31 ± 2.3 ^a^
SMPs group	7.00 ± 1.00 *	13.33 ± 1.15 ^a^

Note: Dislocation time refers to the time when the black sputum (as shown in Figure 2A) of the wound surface is completely removed. It is one of the indicators in the process of epithelialization. Different and same letters within the same row indicate significant (*p* < 0.05) and insignificant different (*p* > 0.05), respectively. * indicates a very significant difference (*p* < 0.01).

**Table 5 molecules-24-04231-t005:** Effect of SMPs on the growth factors of two kinds of growth factors on the 7th day (mean ± S.D., ng/mL prot).

Group	TGF-β1	CCND1
Control	3.33 ± 0.76 ^b^	5.76 ± 1.60 ^a^
Positive Control	11.00 ± 6.81 ^ab^	8.52 ± 1.88 ^a^
SMPs	16.82 ± 6.52 ^a^	7.96 ± 0.68 ^a^

Different and same letters within the same row indicate significant (*p < 0.05*) and insignificant different (*p* > 0.05), respectively.

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
