# Peer review of "Evaluation of Small Molecular Polypeptides from the Mantle of Pinctada Martensii on Promoting Skin Wound Healing in Mice"

_molecules, 2019, doi:10.3390/molecules24234231_

Round 1

Reviewer 1 Report

The paper presents interesting results of in vitro and in vivo research on accelerating wound healing by Small Molecular Polypeptides from Pinctada Martensii. The authors also try to find an explanation for the observed changes in wound healing by assessing coagulation, inflammatory markers and antibacterial activity.

However, there are some minor issues that requires clarification:

Please discuss results of SMPs (composition, characteristic) with current (2019) literature on transcriptomic, proteomic and metabolic analysis of Pinctada Martensii. Please add discussion on possible influence of SMPs on angiotensinergic system and angiogenesis which are important in wound healing.

Author Response

Dear professor,

We are very grateful to you for your detailed review of our imperfect papers, and for your critical suggestions. We must admit that we have a lot of unclearness about the method of introduction of the article, which has caused you too many obstacles in reviewing the manuscript. The following is our reply based on the comments you gave:

Question 1: Please discuss results of SMPs (composition, characteristic) with current (2019) literature on transcriptomic, proteomic and metabolic analysis of Pinctada Martensii. 

Response 1: Based on your suggestions, we refer to the research results of the related literature "Comparative transcriptomic and proteomic analysis of yellow shell and black shell pearl oysters, Pinctada fucata martensii", which was supplemented and highlighted in the discussion section of the article.

Question 2: Please add discussion on possible influence of SMPs on angiotensinergic system and angiogenesis which are important in wound healing. 

Response 2: Based on your suggestion, we had supplemented the discussion. Since we didn't measure the related indexes of angiotensingergic system, we mainly discussed angiogenesis at lines 270-275.

Finally, we thank you again for your valuable suggestions for our work!

Kind regards,

Faming Yang

November 6, 2019

Reviewer 2 Report

The present manuscript describes (mostly) positive effects of polypeptides from Pinctada Martensii on promoting wound healing. Overall the paper is written fairly well, and the experiments appear to be rigorous if somewhat routine. I have a few minor concerns, as follows:

(1) Some editing is required. There are mistakes in various places, such as "growth factor-beta" rather than "transforming growth factor-beta" in the abstract. Another example is an incomplete sentence in section 2.2, lines 81-83. The entire manuscript would benefit from a careful proofread.

(2) Table 3 appears to show an anti-coagulant effect of SMPs, as clotting time is lengthened. The presentation of this finding in the results and the discussion is unclear.

(3) Table 3, the dose column is unnecessary. Additionally, the significance markers are confusing and should be changed to an a, b, c format similar to other figures/tables.

(4) Table 4 is unclear and probably unnecessary.

(5) Figure 2, it is not clear how wound size was measured. Was this done by photoplanimetry (i.e. using the photographs), or directly on the mouse? If the former, was a reference dot used to control for variability in camera placement?

(6) Figure 2, "rate" is probably not the correct description for the outcome being measured. Rather, this is just % closure at each time point.

(7) Table 5, the mixed use of * and a, b, c for significance indicators is confusing. This applies to other tables as well. Additionally, the term "extremely significant" is specious.

(8) Figure 6, the significance indicators in panel B do not make sense.

(9) Discussion, limitations should be presented.

Author Response

Dear professor,

We are very grateful to you for your detailed review of our imperfect papers, and for your critical suggestions and shortcomings. We must admit that we have a lot of unclearness about the method of introduction of the article, which has caused you too many obstacles in reviewing the manuscript. The following will be my Response based on the comments you gave:

Question 1: Some editing is required. There are mistakes in various places, such as "growth factor-beta" rather than "transforming growth factor-beta" in the abstract. Another example is an incomplete sentence in section 2.2, lines 81-83. The entire manuscript would benefit from a careful proofread.

Response 1: Thank you for pointing out the negligence and deficiencies of our work, we have checked and modified the language and grammar of the article.

Question 2: Table 3 appears to show an anti-coagulant effect of SMPs, as clotting time is lengthened. The presentation of this finding in the results and the discussion is unclear.

Response 2: We have re-expressed and supplemented the results and discussion of the anticoagulant effects of SMPs in lines 91-94 and 242-245. Please continue to make suggestions!

Question 3: Table 3, the dose column is unnecessary. Additionally, the significance markers are confusing and should be changed to an a, b, c format similar to other figures/tables.

Response 3: Based on your suggestion, we removed the dose column from Table 3. In addition, we have standardized the expression of the significance markers in the article, lines 98-99 and 125-127.

Question 4: Table 4 is unclear and probably unnecessary.

Response 4: Based on your suggestion, we removed Table 4 and added the content to lines 103-106.

Question 5: Figure 2, it is not clear how wound size was measured. Was this done by photoplanimetry (i.e. using the photographs), or directly on the mouse? If the former, was a reference dot used to control for variability in camera placement?

Response 5: At each time point, we measure the maximum diameter of the wound by vernier calipers, and also have the objectivity of the study, which can be used as a reference for the effect of SMPs on wound closure.

Question 6: Figure 2, "rate" is probably not the correct description for the outcome being measured. Rather, this is just % closure at each time point.

Response 6: We have adopted your suggestion to modify the wound healing rate to the % of wound closure.

Question 7: Table 5, the mixed use of * and a, b, c for significance indicators is confusing. This applies to other tables as well. Additionally, the term "extremely significant" is specious.

Response 7: We have standardized the expression of the significance markers in the article.

Question 8: Figure 6, the significance indicators in panel B do not make sense.

Response 8: Whether this is because of the problem of histogram color, which is not conducive to the range of error lines, we adjust the color of histogram. If our plan is not right, please continue to give more suggestions.

Question 9: Discussion, limitations should be presented.

Response 9: In the discussion, lines 248-249 and 309-312, we supplement the limitations of our research.

Finally, we thank you again for your valuable suggestions for our work!

 Kind regards,

Faming Yang

November 6, 2019

Reviewer 3 Report

The paper by Yang et al. entitled "Evaluation of small molecular polypeptides from the mantle of pinctada martensii on promoting skin wound healing in mice" is a research article that examins the effec of SMPs of the mantle of pinctada martensii on wound healing. The authors found the SMPs promotes wound healing of the skin in mice. Overall, the paper is well written and organized. The paper will be of interest for Readers of the Journal.

I have no major concern but would like to raise the following minor concern.

1. Please state in the Material and Method section how the authors give the SMPs to mice (intravenous? topical use? or oral administration?).

Author Response

Dear professor,

We are very grateful to you for your detailed review of our imperfect papers, and for your critical suggestions. We must admit that we have a lot of unclearness about the method of introduction of the article, which has caused you too many obstacles in reviewing the manuscript. The following is our reply based on the comments you gave:

  Question 1: Please state in the Material and Method section how the authors give the SMPs to mice (intravenous? topical use? or oral administration?).

Response 1: we give SMPs with topical use to mice in this article, and we have supplemented and highlighted the material method section as per your suggestion.

Finally, we thank you again for your valuable suggestions for our work!

Kind regards,

Faming Yang

November 5, 2019